# Kolmogorov-Arnold Networks with Variable Function Basis

## Abstract

Neural networks exhibit exceptional performance in processing complex data, yet their internal structures remain largely unexplored. The emergence of Kolmogorov-Arnold Networks (KANs) represents a significant departure from traditional Multi-Layer Perceptrons (MLPs). In contrast to MLPs, KANs replace fixed activation functions at nodes ("neurons") with learnable activation functions on edges ("weights"), enhancing both accuracy and interpretability. As data evolves, the demand for models that are both flexible and robust minimizing the influence of input data variability continues to grow. Addressing this need, we propose a general framework for KANs utilizing a **V**ariety **B**ernstei**n** Polynomial Function Basis for **K**olmogorov-**A**rnold **N**etworks (VBn-KAN). This framework leverages the Weierstrass approximation theorem to extend function basis within KANs in theory, specifically selecting Bernstein polynomials ($B_n$) for their robustness, assured by the uniform convergence proposition. Additionally, to enhance flexibility, we implement techniques to vary the function basis $B_n$ when handling diverse datasets. Comprehensive experiments across three fields: multivariate time series forecasting, computer vision, and function approximation—demonstrate that our method outperforms conventional approaches and other variants of KANs.

## 1 Introduction

Since the 1990s, the advancement of neural network technology markedly improves the efficiency of CPU program execution, particularly in complex computational tasks and large-scale data processing (Oh and Jung, 2004; Nurvitadhi et al., 2016). This rapid evolution significantly enhances processing speeds, enabling the implementation of more complex models in practical applications (Nguyen and Widrow, 1990a;b; Sze et al., 2017). However, despite these substantial performance improvements, the "black box" nature of neural networks continues to pose challenges, limiting the interpretability and transparency of models (Agarwal et al., 2021; Zhang et al., 2021; Zhang and Zhu, 2018).

In response to these challenges, many researchers recently attempt to conduct in-depth analyses of neural network interpretability from various perspectives. However, providing a comprehensive explanation of neural networks remains daunting, largely due to their complex nonlinear structures and vast parameter spaces. These factors make understanding the internal workings of neural networks exceptionally difficult (Zhang et al., 2018; Lee et al., 2021; Youssef et al., 2023). In this context, the Multi-Layer Perceptron (MLPs), based on the universal approximation theorem (Funahashi, 1989), emerges as a classical neural network structure. It has become an important platform for research due to its relatively simple design and clear hierarchical structure (Borghi et al., 2021; Sharma and Kukreja, 2022; Alnuaim et al., 2022; Rana et al., 2018). The advantages of MLPs networks not only enhance the transparency of the model but also lay the groundwork for further optimization and application.

Although MLPs provide a certain degree of intuitive understanding of neural network structures, their interpretability remains limited. This limited interpretability arises from the fixed activation functions and linear weight structures of MLPs, which obscure the specific contributions of each neuron to the final decision. In Liu et al. (2024), a new architectural approach named "KAN" based on the Kolmogorov-Arnold representation (K-A) theorem (Kolmogorov, 1957b) is proposed

to enhance the interpretability of neural networks. Unlike MLPs, KANs entirely eliminate linear weight matrices; each weight parameter is replaced by a learnable 1D function. Furthermore, they generalize the original K-A representation to arbitrary widths and depths, thereby revitalizing and contextualizing it within the modern deep learning framework.

A variety of KANs exist (Hou et al., 2024), each adapts to different areas (Muyuzhierchengse, 2024; Khochawongwat, 2024; Qiu et al., 2024; Wang et al., 2024a; Koenig et al., 2024; Wang et al., 2024b; De Carlo et al., 2024), enhancing the original KANs for specific downstream tasks. However, some of these models maintain a relatively fixed form, and the principles underlying their mathematical reasoning improvements are not systematically examined. As data evolves, issues such as noise and integrity seem to influence the results of models. Relying solely on manual methods, such as labeling and data cleaning, can consume significant resources. Therefore, reducing the algorithms' dependence on data quality is particularly crucial.

In this paper, we propose KANs with a variety function basis, the: **V**ariety **B**ernstei**n** Polynomial Function Basis for **K**olmogorov-**A**rnold **N**etworks (VBn-KAN). This framework allows the model to adapt to various data types and provides more flexible forms of KANs. Specifically, the Weierstrass approximation theorem (WEIERSTRASS, 1885) broadens the selection of function basis, establishing that polynomials can approximate any continuous function. Based on this theorem, we select the $B_n$ polynomial for its uniform convergence, which ensures robustness in the basis during processes like feature learning, particularly under conditions of uneven distributions or high noise levels. To further enhance the adaptability and flexibility of the model across different scenes, a variety of function bases can effectively meet these challenges. To accomplish this, we utilize versatile Bayesian optimization, leveraging low-dimensional, continuous functions for global optimization, thereby balancing accuracy with exploration.

Our main contributions are summarized as follows:

- To the best of our knowledge, this is the first proposal of a general Kolmogorov-Arnold Network (KANs) with a variable function basis, presented at a mathematical level.
- We employ the Bernstein polynomial as the basis function for KANs, which ensures the model's robustness.
- To enhance the model's flexibility, we utilize a dynamically learnable function basis.

The results of experiments in three fields: multivariate time series forecasting, image classification, and function approximation, further demonstrate the flexibility and robustness of our method.

## 2 RELATED WORK

**Kolmogorov-Arnold Networks (KAN)** KAN (Liu et al., 2024) is inspired by the mathematical foundations established by the Kolmogorov-Arnold representation theorem Kolmogorov (1961; 1957c); Braun and Griebel (2009). These networks offer a promising alternative to Multi-Layer Perceptrons (MLPs). KANs feature learnable activation functions on edges ("weights"), which utilize trainable 1D B-spline functions to process incoming signals. There exists a variety of KANs that select and apply different basis functions for various downstream tasks, such as Bozorgasl and Chen (2024); Muyuzhierchengse (2024); Aghaei (2024); SynodicMonth (2024b); Zhang and Zhang (2024); Azam and Akhtar (2024); Vaca-Rubio et al. (2024); Lai et al. (2024). While these variations achieve good results in specific tasks, they often lack interpretability in their developments.

**Function Variety Basis** Variable basis learning plays a important role in the domains of machine learning and statistical modeling, boasting a long history of research and broad applications across various algorithms and models. Early support vector machines (SVMs) (Boser et al., 1992) leveraged reproducing kernel Hilbert spaces (RKHS) (Berlinet and Thomas-Agnan, 2011) to map data into high-dimensional spaces through kernel techniques, achieving linear separability in otherwise linearly inseparable problems. This approach primarily involved the selection of fixed basis functions. Although effective in certain contexts, it restricts the model's adaptability to complex data structures. Boosting (Freund and Schapire, 1997), a robust ensemble learning technique, enhances a model's predictive performance by iteratively training weak learners and amalgamating them into a strong learner, proving particularly effective in non-parametric learning scenarios. Furthermore,

with the advancement of deep neural networks, the study in Han et al. (2021) introduces the concept of variable parameters and dynamic adjustments to network architecture, offering a versatile method to tailor models to specific data features. These advancements confer numerous benefits, including enhanced efficiency (Bertinetto et al., 2016; Huang et al., 2017a; Lin et al., 2017), increased representation power (Yang et al., 2019; Chen et al., 2020), and improved adaptiveness, compatibility, and generality (He et al., 2016; Kingma, 2014; Liu et al., 2018; Huang et al., 2017b; Yang et al., 2020a), as well as bolstered interpretabilit (Hubel and Wiesel, 1962; Yang et al., 2020b).

**Realizing for Variety Basis**    With the evolution of deep learning and machine learning, the concept of variable basis functions (Yao et al., 2021; Mendel, 2019) has been introduced into various intelligent algorithms, showcasing distinct advantages in addressing complex data structures and nonlinear problems. In deep learning, this concept is operationalized through structured network designs, such as using variable filters in convolutional neural networks (Huang et al., 2021; Gama et al., 2020) or employing conditional parameterization techniques like conditional batch normalization (Wang et al., 2020). In the realm of ensemble learning, variety basis functions enhance predictive performance by constructing and integrating multiple models (base learners) through techniques like bagging (Gu et al., 2018) and the hedge algorithm (Chaudhuri et al., 2009). In reinforcement learning, these functions facilitate strategy optimization challenges, exemplified by strategic gradient methods (Wang et al., 2019) and deep Q-networks (Gu et al., 2016; Lobel et al., 2023).

## 3    METHODOLOGY

Our model, the **V**ariety **B**ernstei**n** Polynomial Function Basis for **K**olmogorov-**A**rnold **N**etworks (VBn-KAN), comprises two core components: the rationale for choosing Bernstein polynomials and the realization of their variety function basis. In this section, we articulate our approach from three perspectives: theoretical foundations, advantages & disadvantages and specific implementations.

### 3.1    THEORETICAL: FOUNDATIONS FOR VARIETY $B_n$ FUNCTION BASES

In the theory of uniform convergence, can polynomials be used to approximate any given continuous function with any desired precision? Reference WEIERSTRASS (1885) provides affirmative answers to this question. Consequently, we present the conclusion of the following well-known theorem:

**Theorem 3.1** *(**Weierstrass Approximation Theorem**) For any $f(x) \in C[a, b]$ and for any $\epsilon > 0$, there exits an algebraic polynomial of the form*

$$p(x) = c_0 + c_1 x + \cdots + c_n x^n, \quad a \leqslant x \leqslant b,$$

*with finite degree $n$ such that the bound*

$$\|f(x) - p(x)\|_{L^\infty} < \epsilon$$

*is satisfied.*

Theorem 3.1 implies that any continuous function on a closed interval can be uniformly approximated by a polynomial function with arbitrary accuracy. This can be viewed as an extension of the theorem in Taylor (1717) to arbitrary continuous functions and as an expansion of the concept in Fourier (1808) within the context of non-periodic basis functions.

The discussions provide insights into univariate functions. Hilbert's 13th problem (Hilbert, 1970) famously posits the impossibility of solving general seventh-degree equations using only functions of two variables. Subsequent research by Kolmogorov (1957a) demonstrates that any function involving multiple variables can be represented with a finite number of three-variable functions. Further studies building on this research, as detailed by Arnol'd (1959), have established that functions of just two variables are sufficient. More precisely:

**Theorem 3.2** *(**Kolmogorov-Arnold representation Theorem**) Any continuous multivariate function $f : [0, 1]^n \to \mathbb{R}$ can be written as univariate functions and using univariate functions in the following form:*

$$f(x) = f(x_1, \cdots, x_n) = \sum_{j=1}^{2n+1} \Phi_j \left[ \sum_{i=1}^{n} \phi_{ij}(x_i) \right], \tag{3.1}$$

*where $\phi_{ij} : [0,1] \to \mathbb{R}$ and $\Phi_j : \mathbb{R} \to \mathbb{R}$.*

Theorem 3.2 demonstrates that a multivariate function can be expressed as a sum of univariate functions. This insight is crucial for understanding the expressive capabilities of complex models, such as neural networks. In Liu et al. (2024), the authors introduce Kolmogorov-Arnold Networks (KANs), an application of neural networks based on Theorem 3.2. Unlike Multi-Layer Perceptrons (MLPs) that are founded on the universal approximation theorem (Funahashi, 1989; Cybenko, 1989; Hornik et al., 1989; Hornik, 1991),KANs feature learnable activation functions on edges, referred to as "neurons" and fix activation functions at nodes, termed "weights". Each weight in KANs is replaced by a univariate function, parametrized as a spline, which means the network contains no linear weights at all.

Specifically, a key feature of optimizing KANs is the implementation of a learnable activation function, represented by $\phi(x)$ in Equation (3.1):

$$\phi(x) = w_b b(x) + w_s \psi(x), \tag{3.2}$$

where

$$\psi(x) = \text{spline}(x), \quad b(x) = \text{silu}(x) = x/(1 + e^{-x}).$$

According to Theorem 3.2, $\phi(x) \in C(\mathbb{R})$, suggesting that learning a high-dimensional function can be effectively approximated by 1D functions. Furthermore, Theorem 3.1 establishes that continuous functions can be approximated by polynomials, allowing the function basis $\psi(x)$ to be extended to any polynomial form. In this context, we utilize Bernstein polynomials ($B_n$) as a variety function basis to replace $\psi(x)$ in Equation (3.2):

$$\widetilde{\phi(x)} = w_b b(x) + w_{B_n} B_n(x),$$

where $B_n$ is defined as:

**Definition 3.1** *(**Bernstein polynomial**) Consider $f(x) \in C[0,1]$, the $n$-th $B_n$ polynomials of $f(x)$ is specified by:*

$$B_n(f) = B_n(f;x) = \sum_{k=0}^{n} f\left(\frac{k}{n}\right) \binom{n}{k} x^k (1-x)^{n-k}, \quad x \in [0,1]. \tag{3.3}$$

To clearly illustrate the distinctions between the structure of the original KAN and our VBn-KAN, we use Figure 1 to present them.

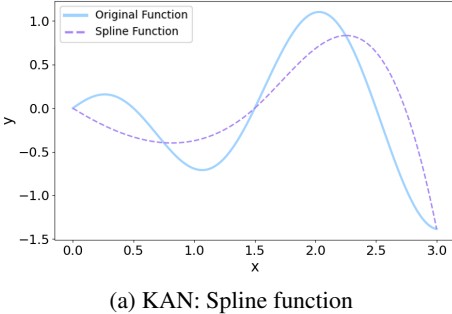 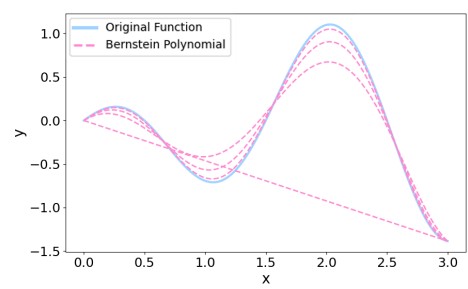

(a) KAN: Spline function        (b) VBn-KAN: Variable degree of $B_n$ polynomial

Figure 1: Comparison of the original KAN (1a) and ours (1b) VBn-KAN structures. The "Original Function" is defined as $f(x) = \cos(\pi x) \cdot \log(1 + |x|)$, which is smooth but not linear. The key difference between (1a) and (1b) lies in the choice of $\psi(x)$ in Equation (3.2), with the former using a spline function and the latter a $B_n$ polynomial. Comparative results indicate that a higher degree of the $B_n$ polynomial yields better approximation results than the spline function. The dashed line in (1b) illustrates varying degrees, where higher degrees more closely approximate the original function.

This change raises three questions:

1) Can polynomials be used to approximate any given continuous function $\phi(x)$?

2) Within the set of polynomials $\mathcal{P}_n$, why choose Bernstein polynomials ($B_n$)?

3) After determining $B_n$ as the new function basis, why introduce variety?

The answer to question 1) is affirmed by Theorem 3.1. We now address the remaining questions.

For question 2), the $B_n$ (Bernstein, 1912) serves as a classical constructive proof method for Theorem 3.1. The main advantage of the Bernstein approximation over Lagrange interpolation (Sauer and Xu, 1995) is highlighted in the following Proposition.

**Proposition 3.1** *For all functions $f$, the sequence $\{B_n f : n = 1, 2, 3, \cdots\}$ converges uniform [1] to $f$ as $B_n (f) \rightrightarrows f(x)$, where $B_n$ is defined by Equation* (3.3).

Similar to Proposition 3.1, the derivative of $B_n$ exhibits the same properties, with $B'_n (f) \rightrightarrows f'(x)$ where $f' \in C[0, 1]$. These uniform convergence properties suggest that as the depth or width of the network increases, $B_n$ polynomials can more uniformly approximate the objective function. This enhances uniformity significantly contributes to improved model generalization.

The uniform convergence of $B_n$ and its derivatives serves as a fundamental concept, best exemplified when approximating a continuous function. This property enables the polynomial to uniformly approximate the objective function across the entire interval, ensuring global consistency and accuracy in the approximation, as depicted in Figure 2. Such uniformity offers distinct advantages in various applications. For instance, in time series forecasting, it enables the network to more effectively capture global trends and seasonal patterns, rather than merely responding to local or short-term fluctuations. In the realm of image classification, it facilitates more balanced learning across different categories, which is particularly beneficial in scenarios with ambiguous category boundaries or imbalanced sample distributions.

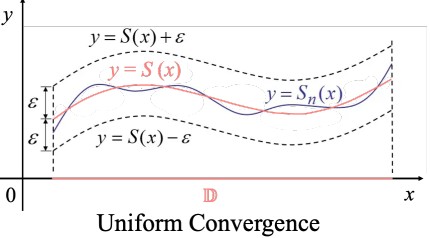 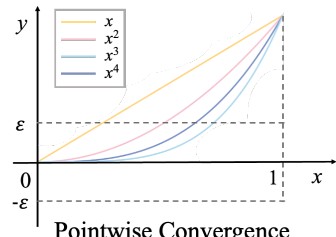

Uniform Convergence          Pointwise Convergence

Figure 2: The distinction between uniform convergence (left) and pointwise convergence (right). In the left figure, $f_n$ consistently remains within the "$\epsilon$-band" across the entire domain, illustrating how uniform convergence maintains the overall structure and continuity of the function. In contrast, the right figure shows that under a given $\epsilon$, regardless of how large $N$ is chosen, there are always $x^n$ values for $n > N$ that do not stay within the $\epsilon$ band. Thus, uniform convergence ensures greater uniformity and robustness.

Moreover, from Equation (3.3), we observe that the smooth (binomial) allocation of weights and the non-negativity of basis functions act to prevent the occurrence of the Runge phenomenon (De Villiers, 2012) during the approximation process.

For question 3), from the previous content, we understand that the approximation error is directly linked to the degree of $B_n$. Specifically, higher degrees tend to yield more accurate approximations, while lower degrees might simplify computation but increase the potential for error. This relationship underlines the need for variety in choosing the appropriate $B_n$ function basis based on specific application requirements and computational constraints.

**Proposition 3.2** *Popoviciu (1935) $B_n$ is Bernstein polynomial of $f(x)$, then*

$$|B_n (f) - f(x)| \lesssim \omega \left(\frac{1}{\sqrt{n}}\right), \tag{3.4}$$

---

[1]**Uniform Convergence** For all $\epsilon > 0$, there existing $N \in \mathbb{Z}^+, N = N(\epsilon)$, s.t. when $n \geqslant N$, there is $|f_n (x) - f(x)| < \epsilon$.

*where $\omega$ is modulus of continuity.*

From Formula (3.4), we understand that as $n$ increases, the approximation error decreases, enabling more precise approximations given a sufficiently large $n$. However, constraints such as computer memory and the availability of examples prevent n from becoming infinitely large ($n \neq \infty$). Under these conditions, for a specified error threshold, there exists a minimum feasible $n$.

**Theorem 3.3** *If $\partial\left[B_n\left(f\right)\right] = n$, for all $\epsilon > 0$, given error limited $\delta$ with $0 < \epsilon \leqslant \delta \ll \infty$, there is*

$$n \geqslant max\left|f''\left(\xi\right)\right|/8\delta.$$

The proof of Theorem 3.3 is provided in the Appendix . For a closed interval $[a, b]$ (where $([0, 1] \subsetneq [a, b])$), Theorem 3.3 is also applicable by virtue of Theorem A.1. Thus, theoretically, we can strike a balance between accuracy and memory usage. Consequently, the variable function basis offers a broader solution for feature learning in neural networks.

## 3.2 DISCUSSION: ADVANTAGES AND LIMITATIONS OF VARIETY FUNCTION BASIS $B_n$

In existing KAN methods, the function basis and its parameters are not adaptive (Muyuzhierchengse, 2024; SynodicMonth, 2024b; Khochawongwat, 2024; Li, 2024a). Variable function basis, however, play a crucial role in KANs by allowing the network to enhance the model's representational power. This enhancement is achieved by adaptively selecting basis functions that are best suited for specific tasks.

- Data Features Perspective: The model can adapt to the non-stationary and multi-scale features of different data types. For instance, in multivariate time-series data forecasting, it effectively captures both local and global features and better processes sudden events. In image classification, the model can adjust its form and parameters based on specific features such as edge density and color distribution, thereby better accommodating the characteristics of various image types.

- Model Training Perspective: When dealing with non-stationary or high-noise data, choosing a higher degree enhances the model's adaptability and accuracy; conversely, employing lower frequencies reduces computational complexity and helps prevent overfitting. Additionally, the uniform convergence proposition minimizes the model's data dependence by maintaining a fixed interval $\varepsilon$ with sufficient $N$, thus striking a balance between error and computational costs.

However, employing a variety function basis might require more time and space than a fixed basis. The introduction of variable basis functions enables the KAN model to adjust adaptively based on data features, thus improving prediction accuracy, enhancing generalization to new data, and effectively reducing the risk of overfitting. This enhanced adaptability not only boosts model performance but also facilitates more efficient resource allocation, making it particularly suitable for complex application scenarios that demand high performance and real-time responses. Therefore, although the initial resource consumption may be high, the investment proves cost-effective in the long run, especially in dynamic and evolving data environments.

## 3.3 REALIZATION: ACHIEVING VARIETY IN FUNCTION BASIS

The selection of the $B_n$ degree is a non-convex and combinatorial optimization problem, primarily because the relationship between approximation error and $B_n$ degree is non-monotonic; it can exhibit non-smooth or discontinuous variations with changes in degree. Consequently, this scene may present multiple local optimal solutions. For instance, under certain data distributions and objective function configurations, both lower and higher iterations might achieve comparable error levels, while some intermediate iterations might result in higher errors. These local optima complicate the search for the global optimum. In this situation, there is a need to balance minimizing error (as discussed in Proposition 3.2) and adhering to computational resource constraints (as outlined in Theorem 3.3).

To determine the degree of the $B_n$ basis, several adaptive models such as additive structures (Hastie, 2017), Multi-armed Bandits (Burtini et al., 2015), and deep hedging (Chaudhuri et al., 2009) are

available. However, these models are typically employed for multivariate impact analysis or for combining and selecting multiple models. In this context, we employ Bayesian Optimization (BO) to address the aforementioned challenges. BO effectively navigates non-convex spaces to identify the global optimum by constructing a probabilistic model, typically a Gaussian process, for the objective function, such as approximation error. Furthermore, it automatically balances exploration and exploitation (Garnett, 2023). Notably, traditional BO is a leading method in expensive black-box optimization due to its data efficiency. A comprehensive and updated survey about BO is provided in Wang et al. (2023).

To further optimize the variety of $B_n$ polynomial bases, we employ the versatile Bayesian Optimization tool SMAC3 (Lindauer et al., 2022) for its robustness and flexibility. SMAC3 is particularly suited for low-dimensional and continuous functions, aligning with our objectives. Specifically, the variety basis $n$ is treated as a hyperparameter optimization problem. We use Mean Absolute Error (MAE) as an example metric, consistent with other metrics discussed in Section 4.2. We define the cost function $L$ as

$$L\left(D_{\text{train}}, D_{\text{val}}; n\right) = \frac{1}{|D_{\text{val}}|} \sum_{(x,y) \in D_{\text{val}}} \left[y - g\left(x; n\right)\right]^2, \tag{3.5}$$

The cost function $L$ is defined as follows, where $D_{\text{train}}$ is the training dataset, $D_{\text{val}}$ is the validation dataset, $g(x; n)$ is the predicted output of the model under the hyperparameters $n$, and $y$ represents the actual value (ground truth). During the optimization process, SMAC3 experiments with different $n$ values and calculates the corresponding $L$ values. By constructing a Gaussian process model, SMAC3 predicts the performance of the cost function for unexplored $n$ values. It employs the acquisition function, Expected Improvement (EI), to select the subsequent n value that is most likely to enhance the objective function. EI is defined as

$$\text{EI}(n) = \mathbb{E}[\max(0, \mu(n) - g(x_{\text{best}}))],$$

where $\mu\left(n\right)$ is the predicted mean of $L$ under hyperparameter basis $n$, and $g(x_{\text{best}})$ represents the currently observed lowest value of $L$.

# 4 EXPERIMENTS

## 4.1 PROBLEM FORMULATION

**Multivariate Time Series (MTS) Forecasting**  In MTS forecasting, consider historical observations represented as $X_{\text{his}} = \{X_1, \cdots, X_T\} \in \mathbb{R}^{T \times N}$, where each observation $X_*$ consists of $N$ variates. The objective is to predict the future $L$ time steps $X_{\text{fut}} = \{X_{T+1}, \cdots, X_{T+L}\} \in \mathbb{R}^{L \times N}$. For notational convenience, $X_t$ denotes the set of variates recorded simultaneously at time step $t$, with $x_*^t$ representing the variate indexed by at this time step.

**Image Classification**  In image classification, the dataset $\mathcal{D}$ consists of pairs of images and their corresponding labels. Each image $X_i$ is represented as a tensor in $\mathbb{R}^{H \times W \times C}$, with $H, W$ and $C$ denoting the height, width and number of channels, respectively. The objective is to develop a function $f : \mathbb{R}^{H \times W \times C} \to \mathbb{Y}$ that maps each image to a set of labels $\mathbb{Y}$, with the aim of minimizing the discrepancy between the predicted labels $\widehat{y} = f(X)$ and the actual labels across the dataset $\mathcal{D}$.

**Function Approximation**  The function that this paper seeks to approximate is a multivariate constant function. For a given function $f\left(x_1, \cdots, x_n\right) \in \mathbb{R}^n$, our objective is to generate a function $g\left(x_1, \cdots, x_n\right)$ such that the approximation error is significantly reduced.

## 4.2 FUNDAMENTAL INFORMATION

We evaluate our model in three distinct fields: MTS forecasting, CV and function approximation. We compare its performance against corresponding baseline methods as well as various types of KANs.

**Datasets**  In MTS forecasting, we conduct experiments on the ETT(h1, h2, m1, m2) dataset (Zhou et al., 2021), the ECL dataset (Trindade, 2015), and the Weather dataset (for Environmental Information, 2013). For image classification, we evaluate our model on the MNIST dataset (LeCun et al.,

1998), CIFAR-10, CIFAR-100 (Krizhevsky et al., 2009), and Fashion-MNIST (Xiao et al., 2017). For function approximation, we utilize the functions discussed in Section 4.1 of Liu et al. (2024).

**Baseline** We select a range of baseline models, including basic models and various KANs, within their respective fields. The basic models include RLinear (Li et al., 2023), DLinear (Zeng et al., 2023), SCINet (Liu et al., 2022), and FEDformer (Zhou et al., 2022). For the variety of KANs, we consider WavKAN (Bozorgasl and Chen, 2024), TaylorKAN (Muyuzhierchengse, 2024), JacobKAN (Aghaei, 2024), FourierKAN (Xu et al., 2024), ConvKAN (StarostinV, 2024), ChebyKAN (SynodicMonth, 2024a), as well as FastKAN and RBFKAN (Li, 2024b).

**Metric** For the MTS task, we utilize the metrics as specified in Zhou et al. (2021), namely Mean Squared Error (MSE) and Mean Absolute Error (MAE). For the image classification, we adopt Accuracy (Acc) (Hussain et al., 2019) as our metric. For function approximation, we use Average Displacement Error (ADE) to quantify the error, as it directly describes the displacement in space (Zhang et al., 1988).

### 4.3 RESULTS

Here, we present the results of three types as described in Section 4.1. In this section, the **best** result is highlighted in bold red, while the second best result is underlined in blue.

Regarding the results of MTS forecasting, we provide the average (ave) results in Table 1. For the whole predicted length, please refer to Appendix B.

Table 1: The results for MTS forecasting (above the double horizontal line are the basic methods in MTS forecasting, while below the line are the methods based on KANs)

| Dataset | VBn-KAN | | RLinear | | DLinear | | SCINet | | FEDformer | |
|---|---|---|---|---|---|---|---|---|---|---|
| | MSE | MAE | MSE | MAE | MSE | MAE | MSE | MAE | MSE | MAE |
| ETTm1 | **0.397** | 0.400 | 0.414 | 0.407 | 0.403 | 0.407 | 0.485 | 0.481 | 0.448 | 0.452 |
| ETTm2 | **0.282** | 0.331 | 0.286 | 0.327 | 0.350 | 0.401 | 0.571 | 0.537 | 0.305 | 0.349 |
| ETTh1 | **0.435** | **0.431** | 0.446 | 0.434 | 0.456 | 0.452 | 0.747 | 0.647 | 0.440 | 0.460 |
| ETTh2 | 0.397 | 0.414 | **0.374** | **0.398** | 0.559 | 0.515 | 0.954 | 0.723 | 0.437 | 0.449 |
| ECL | 0.217 | **0.297** | 0.219 | 0.298 | **0.212** | 0.300 | 0.268 | 0.365 | 0.214 | 0.327 |
| Weather | **0.262** | **0.289** | 0.272 | 0.291 | 0.265 | 0.317 | 0.292 | 0.363 | 0.309 | 0.360 |

| Dataset | KAN | | FourierKAN | | WavKAN | | TaylorKAN | | JacobKAN | |
|---|---|---|---|---|---|---|---|---|---|---|
| | MSE | MAE | MSE | MAE | MSE | MAE | MSE | MAE | MSE | MAE |
| ETTm1 | 0.421 | 0.413 | 0.709 | 0.550 | **0.397** | **0.399** | 0.402 | 0.400 | 0.404 | 0.405 |
| ETTm2 | 0.304 | 0.351 | 0.328 | 0.363 | 0.287 | 0.333 | 0.285 | 0.329 | 0.284 | 0.331 |
| ETTh1 | 0.472 | 0.453 | 0.572 | 0.525 | 0.444 | 0.435 | 0.445 | **0.431** | **0.435** | 0.440 |
| ETTh2 | 0.429 | 0.436 | 0.475 | 0.462 | 0.417 | 0.424 | 0.392 | 0.411 | 0.397 | 0.414 |
| ECL | 0.223 | 0.304 | 0.222 | 0.301 | 0.221 | 0.300 | 0.224 | 0.300 | 0.220 | 0.301 |
| Weather | 0.273 | 0.298 | 0.287 | 0.313 | 0.264 | 0.291 | 0.269 | 0.294 | 0.266 | 0.292 |

The experimental results demonstrate that VBn-KAN exhibits significant advantages on most datasets, particularly on the ETTm1, ETTm2, and ETTh2 datasets. Specifically, on the ETTm1 dataset, VBn-KAN achieves an MSE of 0.397, representing a substantial 44.0% performance improvement compared to Fourier KAN's MSE of 0.709. For the ETTm2 dataset, VBn-KAN's MSE of 0.282 signifies a 19.4% performance enhancement over DLlinear's MSE of 0.350 (the lowest MSE for comparison). Across all considered datasets, VBn-KAN demonstrates an average MSE improvement of approximately 20% to 25% compared to DLlinear. Notably, VBn-KAN's most significant advantage is observed on the Weather dataset, where it achieves a MAE of 0.289, marking a 6.5% improvement over FEDformer's MAE of 0.309 (the best performance among all KAN variants). When compared to the traditional SCINet, which has a MAE of 0.363, VBn-KAN's improvement is as high as 20.4%.

For image classification, the accuracy results are presented in Table 2.

Table 2: The accuracy results for image classification (the networks listed after "KAN" are based KANs. For brevity, we use the abbreviation of their basis functions and omit the term "KAN")

| Dataset | VBn-KAN | KAN | Conv | Fast | Cheby | ReLU | Faster | Jacobi | RBF |
|---|---|---|---|---|---|---|---|---|---|
| MNIST | **98.49%** | 97.21% | 94.56% | 97.55% | 97.01% | 92.84% | 92.87% | 97.44% | 98.43% |
| cifar-10 | **69.79%** | 64.04% | 62.40% | 63.01% | 49.29% | 52.59% | 65.20% | 44.2% | 63.25% |
| cifar-100 | 39.22% | **43.62%** | 36.27% | 24.98% | 17.66% | 20.30% | 36.67% | 22.38% | 22.83% |
| F-MNIST [†] | 90.45% | 89.77% | - | **90.67%** | 87.73% | 87.27% | 89.59% | 88.24% | 88.71% |

[†] This represent the dataset Fashion-MNIST (Xiao et al., 2017).

On both the MNIST and CIFAR-10 datasets, our VBn-KAN model secures the highest accuracy, surpassing competing algorithms by substantial margins of up to 5.65% and 17.2% respectively. While in the CIFAR-100 dataset, our approach lags slightly by a mere 0.33% behind the top-performing RBFKAN model, it continues to demonstrate strong competitive advantage. In the F-MNIST dataset, although our method is within 2.2% of the leading Conv model, it consistently shows robust performance across diverse benchmarks. Overall, the VBn-KAN model consistently delivers superior or competitive results, underlining its efficacy in various image classification tasks.

Regarding the function approximation, to quantify the error across multiple variables, we utilize the geometric error metric known as the Average Displacement Error (ADE). The results obtained are shown in Table 3.

Table 3: The ADE metric for function approximation

| | VBn-KAN | KAN | Cheby | Fast | Wave | Jacobi | ReLu | Fourier | Taylor | RBF |
|---|---|---|---|---|---|---|---|---|---|---|
| $f_1$ | 0.0087 | **0.0057** | 0.2401 | 0.2055 | 0.1803 | 12.6966 | 0.6025 | 0.2320 | 0.1791 | 94.3770 |
| $f_2$ | **2.4014** | 2.6947 | 2.6739 | 2.6269 | 2.7350 | 14.5713 | 60.4872 | 2.7944 | 2.6075 | 96.3917 |
| $f_3$ | 0.3756 | **0.3665** | 1.3055 | 1.2677 | 0.8166 | 12.0580 | 59.2380 | 0.9249 | 1.3631 | 96.2014 |
| $f_4$ | 0.2047 | **0.0899** | 0.8582 | 0.7981 | 0.9450 | 18.5012 | 52.5171 | 0.6897 | 0.8362 | 0.9626 |
| $f_5$ | **0.0769** | 0.1393 | 0.9292 | 0.8844 | 0.5351 | 260.7600 | 0.8007 | 0.6075 | 0.7993 | 0.5115 |

where $f_1 = xy$, $f_2 = x/y$, $f_3 = \exp\left[J_0\left(20x\right) + y^2\right]$ ($J_0\left(20x\right)$ is Bessel function), $f_4 = \tanh\left[5\left(x_1^4 + x_2^4 + x_3^4 - 1\right)\right]$ and $f_5 = \sqrt{\left(x_1 - x_2\right)^2 + \left(x_3 - x_4\right)^2}$.

In the function approximation experiments involving these five functions, our VBn-KAN method exhibits remarkable performance in approximating $f_2$ and $f_5$, particularly achieving the minimum error in $f_5$. For $f_1$ and $f_3$, the error margins compared to the optimal values are merely 0.003 and 0.0091, respectively. In the case of $f_2$, VBn-KAN significantly outperforms the second-best result KAN, with an approximation accuracy improvement of approximately 10.9%. For $f_5$, VBn-KAN achieves an error of 0.0769, markedly lower than KAN's 0.1393, reducing the error by 0.0624, which corresponds to an enhancement of about 44.8%. This underscores the robust approximation capabilities of the our method in handling distance functions.

Here we use $f_5$ as an example, the error heatmap is illustrate in Figure 3:

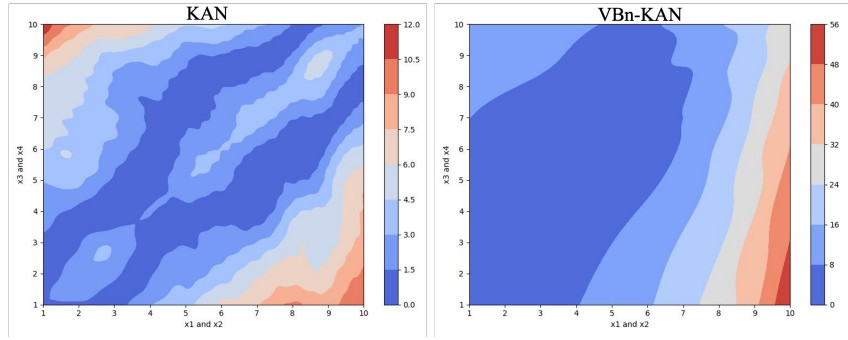

Figure 3: ADE heatmap comparison for function $f_5$ using KAN and VBn-KAN.

In approximating the Euclidean distance function $f_5$, VBn-KAN demonstrates clear advantages over KAN. VBn-KAN's error distribution is smoother, with fewer sharp spikes and a more controlled error increase, particularly in the high $x_1$ and $x_2$ regions. This stability is crucial for a distance-based function, as small fluctuations can significantly impact accuracy. In contrast, KAN shows more irregular error patterns, with larger spikes that indicate less consistent performance. Thus, VBn-KAN offers a more stable and reliable approximation for $f_5$.

### 4.4 ABLATION STUDY

In this section, we verify that the effectiveness of variety function basis $B_n$ in Table 4 and 5, the stability during the approximation process in Table 6.

Table 4: The results of fixed function basis $n$ in MTS forecasting

| D | Length | VBn-KAN MSE | VBn-KAN MAE | $n=1$ MSE | $n=1$ MAE | $n=5$ MSE | $n=5$ MAE | $n=10$ MSE | $n=10$ MAE |
|---|---|---|---|---|---|---|---|---|---|
| ETTm1 | 96 | 0.340 | **0.368** | **0.339** | **0.368** | 0.343 | 0.368 | 0.398 | 0.377 |
| | 192 | 0.378 | **0.382** | **0.377** | 0.385 | 0.382 | 0.387 | 0.382 | 0.392 |
| | 336 | **0.409** | 0.409 | 0.410 | **0.406** | 0.414 | **0.406** | 0.414 | 0.413 |
| | 720 | **0.462** | 0.440 | 0.470 | **0.438** | 0.478 | 0.440 | 0.465 | 0.441 |
| | avg | **0.397** | 0.400 | 0.399 | **0.399** | 0.404 | 0.400 | 0.415 | 0.406 |
| ETTh1 | 96 | **0.382** | **0.394** | 0.402 | 0.409 | 0.414 | 0.422 | 0.574 | 0.502 |
| | 192 | **0.426** | **0.424** | 0.444 | 0.434 | 0.467 | 0.457 | 0.673 | 0.545 |
| | 336 | **0.469** | **0.446** | 0.513 | 0.473 | 0.505 | 0.479 | 0.759 | 0.584 |
| | 720 | **0.464** | **0.459** | 0.546 | 0.509 | 0.549 | 0.500 | 0.881 | 0.634 |
| | avg | **0.435** | **0.431** | 0.476 | 0.456 | 0.484 | 0.465 | 0.722 | 0.566 |

Table 5: The fixed $n$ accuracy for image classification

| Datasets | VBn-KAN | $n=1$ | $n=5$ | $n=10$ |
|---|---|---|---|---|
| MNIST | **98.49%** | 95.37% | 94.67% | 89.32% |
| cifar-10 | **69.79%** | 59.04% | 64.10% | 53.17% |
| cifar-100 | **39.22%** | 25.98% | 31.78% | 26.17% |

Table 6: The stability of image classification

| Datasets | VBn-KAN | Conv | Fast | Cheby | ReLU | Faster |
|---|---|---|---|---|---|---|
| cifar-100 | **0.014750** | 0.180616 | 0.086888 | 0.078740 | 0.928559 | 2.736019 |
| MNIST | **0.000218** | 0.028284 | 0.097624 | 0.077603 | 0.260896 | 1.199764 |

Compared with the methods where the function basis $n$ is fixed, it is evident that our variable method significantly outperforms these methods in both the ETTh1 dataset and three CV image classification datasets. Specifically, in ETTh1, our method yields an error that is $0.095$ less than the average error of the three $n$-fixed methods. In the CV tasks, our VBn-KAN exceeds the performance of the listed datasets, achieving maximum accuracy values of $9.17\%$, $16.62\%$, and $13.24\%$ for the three fixed frequency datasets, respectively. Although our model does not perform the best in the ETTm1 dataset among the listed times, for the error values that do not exceed, our results rank second and have an average difference of only $0.002$ from the minimum error in terms of both ADE and FDE.

## 5 CONCLUSION

In this paper, we present a general framework, the **V**ariety **B**ernstei**n** Polynomial Function Basis for **K**olmogorov-**A**rnold **N**etworks (VBn-KAN). This framework extends the range of basis functions in theory and enhances flexibility and robustness through the variety of $B_n$ function bases driven by the uniform convergence proposition. Furthermore, it reduces the influence of data on the model and provides a balanced approach in feature learning by minimizing error while managing computing costs. Experiments in multivariate time series forecasting, image classification, and function approximation demonstrate that our VBn-KAN achieves significant results both theoretically and practically.

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

## A APPENDIX

### A.1 THE PROOF OF THEOREM 3.3

**Theorem 3.3** *For all $\epsilon > 0$, if $\partial [B_n(f)] = n$, for a given error limit $\delta$ with $0 < \epsilon \leqslant \delta \ll \infty$, then*

$$n \geqslant \max|f''^{(\xi)}|/8\delta.$$

*Proof:* First, we calculate the error between $f(x)$ and $B_n(f)$

$$|f(x) - B_n(f)| = \left| \sum_{k=0}^{n} \left[ f(x) - f\left(\frac{k}{n}\right) \right] \binom{n}{k} x^k (1-x)^{n-k} \right|$$

$$\leqslant \left| \sum_{k=0}^{n} \left[ -f''^{\left(\frac{k}{n}-x\right)} - \frac{1}{2} f'' \left(\frac{k}{n} - x\right)^2 \right] \cdot P_B(k) \right| \tag{A.1}$$

$$= \frac{1}{2} f''(\xi) \sum_{k=0}^{n} \left(\frac{n}{k} - x\right)^2 P_B(k) \qquad \xi \in \left(x, \frac{k}{n}\right). \tag{A.2}$$

In Equation (A.1), $P_B(k) \triangleq C_n^k x^k (1-x)^{n-k}$. From Equation (A.2), we next proceed to prove

1. $\sum_k \left(\frac{n}{k} - x\right) \cdot P_B(k) = 0$,
2. $\sum_k \left(\frac{n}{k} - x\right)^2 \cdot P_B(k) = \frac{x}{n}(1-x)$.

For Equation 1, applying the Central Limit Theorem (CLT) as discussed in Billingsley (1995), we consider the total difference of weights, represented by $\left(\frac{n}{k} - x\right)$. Specifically, the probability $p$ satisfies $p = x = k/n$. In this case, as described in Johnson et al. (2005), we have $E(k) = nx$. Thus, we can derive the expectation as follows:

$$E\left(\frac{k}{n}\right) = \frac{E(k)}{n} = x. \tag{A.3}$$

Therefore, Equation 1 simplifies to $E(k/n) - x = 0$.

For Equation 2, we employ a similar method; here $\left(\frac{n}{k} - x\right)^2$ represents the squared difference in weights between $\frac{k}{n}$ and $x$, alternatively described as the deviation between observation and expectation. According to Equation (A.3),

$$D\left(\frac{k}{n}\right) = \frac{nx}{n^2} \cdot (1-x) = \frac{x(1-x)}{n}.$$

Equation 2 corresponds to the squared deviation $\left(\frac{n}{k} - x\right)^2$ based on weights $P_B(k)$. Moreover

$$\because E\left[\left(\frac{k}{n} - x\right)^2\right] = D\left(\frac{k}{n}\right),$$

$$\therefore (A.2) = \frac{1}{2} f''(\xi) \frac{x(1-x)}{n} \leqslant \frac{M_2}{8n}, \text{ with } M_2 = \max_{\xi \in \mathbb{D}} f''(\xi).$$

Considering the error limit $\delta$, we have:

$$\frac{M_2}{8n} < \delta \Rightarrow n \geqslant \frac{M_2}{8\delta}.$$

$\square$

**Theorem A.1** *If $f \in C[a,b]$, $B_n^{\omega}(f) \in [a,b]$ and $\partial [B_n^{\omega}(f)] = n$. For all $\epsilon > 0$, $0 < \epsilon \leqslant \delta \ll \infty$, $\delta$ is given error limitation, then:*

$$n \geqslant \frac{\max[|f''(\widetilde{\xi}) \cdot \omega|] \cdot (b-a)^2}{8\delta},$$

*where $\omega$ is the weights of weighted Bernstein polynomials $B_n^{\omega}(f)$.*

*Proof:*   Here the error is $L^\infty$ norm, the definition of $B_n^\omega(f)$ is

$$B_n^\omega(f) = \sum_{k=0}^{n} f\left(\frac{k}{n}\right) \omega\left(\frac{k}{n}\right) \binom{n}{k} x^k (1-x)^{n-k}$$

Let $u = (x-a)/(b-a)$, then $u \in [0,1]$. So $\widetilde{M_2}$ is similar to Theorem 3.3, here

$$\widetilde{M_2} = \max_{u\in[0,1]} |g''(u)| = (b-a)^2 \left|f''(\xi)\right|,$$

where $g(u) = f\left(\frac{u-a}{b-a}\right)$. The next following prove is similar to Theorem 3.3. □

# B  THE WHOLE PREDICTED LENGTH RESULTS FOR MULTIVARIETY TIME SERIES (MTS) FORECASTING

In this section, the **best** result is bolded in red while the second one is underline in blue.

Table 7: The whole results for MTS forecasting by basic method.

| D | L
H | VBn-KAN
MSE | MAE | RLinear
MSE | MAE | DLinear
MSE | MAE | SCINet
MSE | MAE | FEDformer
MSE | MAE |
|---|---|---|---|---|---|---|---|---|---|---|---|
| ETTm1 | 96 | **0.340** | **0.368** | 0.355 | 0.376 | 0.345 | 0.372 | 0.418 | 0.438 | 0.379 | 0.419 |
| | 192 | 0.378 | **0.382** | 0.391 | 0.392 | 0.380 | 0.389 | 0.439 | 0.450 | 0.426 | 0.441 |
| | 336 | 0.409 | 0.409 | 0.424 | 0.415 | 0.413 | 0.413 | 0.490 | 0.485 | 0.426 | 0.441 |
| | 720 | **0.462** | 0.440 | 0.487 | 0.450 | 0.474 | 0.453 | 0.595 | 0.550 | 0.543 | 0.490 |
| | avg | **0.397** | **0.400** | 0.414 | 0.407 | 0.403 | 0.407 | 0.485 | 0.481 | 0.448 | 0.452 |
| ETTm2 | 96 | **0.176** | **0.263** | 0.182 | 0.265 | 0.193 | 0.292 | 0.286 | 0.377 | 0.203 | 0.287 |
| | 192 | **0.244** | 0.309 | 0.246 | **0.304** | 0.284 | 0.362 | 0.399 | 0.445 | 0.269 | 0.328 |
| | 336 | **0.304** | 0.348 | 0.307 | **0.342** | 0.369 | 0.427 | 0.637 | 0.591 | 0.325 | 0.366 |
| | 720 | **0.402** | 0.402 | 0.407 | **0.398** | 0.554 | 0.522 | 0.960 | 0.735 | 0.421 | 0.415 |
| | avg | **0.282** | **0.331** | 0.286 | 0.327 | 0.350 | 0.401 | 0.571 | 0.537 | 0.305 | 0.349 |
| ETTh1 | 96 | 0.382 | **0.394** | 0.386 | 0.395 | 0.386 | 0.400 | 0.654 | 0.599 | **0.376** | 0.419 |
| | 192 | 0.426 | **0.424** | 0.437 | **0.424** | 0.437 | 0.432 | 0.719 | 0.631 | **0.420** | 0.448 |
| | 336 | 0.469 | 0.446 | 0.479 | 0.446 | 0.481 | 0.459 | 0.778 | 0.659 | **0.459** | 0.465 |
| | 720 | 0.464 | **0.459** | 0.481 | 0.470 | 0.519 | 0.516 | 0.836 | 0.699 | 0.506 | 0.507 |
| | avg | **0.435** | **0.431** | 0.446 | 0.434 | 0.456 | 0.452 | 0.747 | 0.647 | 0.440 | 0.460 |
| ETTh2 | 96 | 0.300 | 0.344 | **0.288** | **0.338** | 0.333 | 0.387 | 0.707 | 0.621 | 0.358 | 0.397 |
| | 192 | 0.391 | 0.402 | **0.374** | 0.390 | 0.477 | 0.476 | 0.860 | 0.689 | 0.429 | 0.439 |
| | 336 | 0.436 | 0.443 | **0.415** | **0.426** | 0.594 | 0.541 | 1.000 | 0.744 | 0.496 | 0.487 |
| | 720 | 0.463 | 0.466 | **0.420** | **0.440** | 0.831 | 0.657 | 1.249 | 0.838 | 0.463 | 0.474 |
| | avg | **0.397** | **0.414** | 0.374 | 0.398 | 0.559 | 0.515 | 0.954 | 0.723 | 0.437 | 0.449 |
| ECL | 96 | 0.196 | **0.280** | 0.201 | 0.281 | 0.197 | 0.282 | 0.247 | 0.345 | **0.193** | 0.308 |
| | 192 | 0.199 | **0.282** | 0.201 | 0.283 | **0.196** | 0.285 | 0.257 | 0.355 | 0.201 | 0.315 |
| | 336 | 0.215 | **0.296** | 0.215 | 0.298 | **0.209** | 0.301 | 0.269 | 0.369 | 0.214 | 0.329 |
| | 720 | 0.258 | 0.331 | 0.257 | 0.331 | **0.245** | 0.333 | 0.299 | 0.390 | 0.246 | 0.355 |
| | avg | **0.217** | **0.297** | 0.219 | 0.298 | 0.212 | 0.300 | 0.268 | 0.365 | 0.214 | 0.327 |
| Weather | 96 | **0.175** | **0.226** | 0.192 | 0.232 | 0.196 | 0.255 | 0.221 | 0.306 | 0.217 | 0.296 |
| | 192 | **0.225** | 0.270 | 0.240 | 0.271 | 0.237 | 0.296 | 0.261 | 0.340 | 0.276 | 0.336 |
| | 336 | **0.282** | **0.305** | 0.292 | 0.307 | 0.283 | 0.335 | 0.309 | 0.378 | 0.339 | 0.380 |
| | 720 | 0.366 | 0.356 | 0.364 | 0.353 | **0.345** | 0.381 | 0.377 | 0.427 | 0.403 | 0.428 |
| | avg | **0.262** | **0.289** | 0.272 | 0.291 | 0.265 | 0.317 | 0.292 | 0.363 | 0.309 | 0.360 |

Table 8: The whole predicted length results for MTS forecasting by KAN-based method.

| D | ML | VBn-KAN MSE | VBn-KAN MAE | KAN MSE | KAN MAE | FourierKAN MSE | FourierKAN MAE | WavKAN MSE | WavKAN MAE | TaylorKAN MSE | TaylorKAN MAE | JacobKAN MSE | JacobKAN MAE |
|---|---|---|---|---|---|---|---|---|---|---|---|---|---|
| ETTm1 | 96 | **0.340** | **0.368** | 0.358 | 0.379 | 0.705 | 0.541 | 0.341 | 0.370 | 0.340 | 0.368 | 0.353 | 0.376 |
| | 192 | 0.378 | **0.382** | 0.403 | 0.401 | 0.707 | 0.547 | **0.374** | 0.384 | 0.380 | 0.386 | 0.386 | 0.391 |
| | 336 | 0.409 | 0.409 | 0.435 | 0.420 | 0.706 | 0.550 | **0.406** | **0.405** | 0.412 | 0.406 | 0.413 | 0.410 |
| | 720 | **0.462** | 0.440 | 0.490 | 0.450 | 0.718 | 0.561 | 0.465 | **0.438** | 0.474 | 0.439 | 0.465 | 0.442 |
| | avg | **0.397** | 0.400 | 0.421 | 0.413 | 0.709 | 0.550 | **0.397** | **0.399** | 0.402 | 0.400 | 0.404 | 0.405 |
| ETTm2 | 96 | **0.176** | **0.263** | 0.203 | 0.291 | 0.230 | 0.306 | 0.179 | 0.265 | 0.181 | 0.264 | 0.177 | **0.263** |
| | 192 | **0.244** | 0.309 | 0.255 | 0.318 | 0.287 | 0.338 | 0.247 | 0.309 | 0.247 | 0.306 | 0.246 | 0.308 |
| | 336 | **0.304** | 0.348 | 0.329 | 0.370 | 0.349 | 0.377 | 0.311 | 0.349 | 0.308 | 0.345 | 0.309 | 0.348 |
| | 720 | **0.402** | 0.402 | 0.428 | 0.428 | 0.446 | 0.428 | 0.412 | 0.408 | 0.405 | 0.401 | 0.406 | 0.403 |
| | avg | **0.282** | 0.331 | 0.304 | 0.351 | 0.328 | 0.363 | **0.287** | **0.333** | 0.285 | 0.329 | 0.284 | 0.331 |
| ETTh1 | 96 | 0.382 | **0.394** | 0.418 | 0.419 | 0.514 | 0.482 | 0.396 | 0.402 | 0.388 | 0.398 | 0.389 | 0.411 |
| | 192 | 0.426 | **0.424** | 0.467 | 0.447 | 0.573 | 0.522 | 0.439 | 0.430 | 0.438 | 0.424 | 0.434 | 0.437 |
| | 336 | 0.469 | 0.446 | 0.509 | 0.470 | 0.613 | 0.545 | 0.479 | 0.449 | 0.477 | **0.442** | 0.464 | 0.451 |
| | 720 | 0.464 | **0.459** | 0.493 | 0.478 | 0.588 | 0.551 | 0.464 | 0.461 | 0.477 | 0.461 | **0.452** | 0.460 |
| | avg | **0.435** | 0.431 | 0.472 | 0.453 | 0.572 | 0.525 | **0.444** | **0.435** | 0.445 | 0.431 | 0.435 | 0.440 |
| ETTh2 | 96 | 0.300 | 0.344 | 0.327 | 0.367 | 0.411 | 0.420 | 0.321 | 0.358 | 0.297 | 0.344 | 0.310 | 0.352 |
| | 192 | 0.391 | 0.402 | 0.409 | 0.420 | 0.475 | 0.455 | 0.407 | 0.413 | 0.387 | 0.401 | 0.396 | 0.405 |
| | 336 | 0.436 | 0.443 | 0.475 | 0.468 | 0.509 | 0.483 | 0.452 | 0.450 | 0.434 | 0.441 | 0.437 | 0.441 |
| | 720 | 0.463 | 0.466 | 0.506 | 0.490 | 0.505 | 0.490 | 0.487 | 0.477 | 0.448 | 0.458 | 0.445 | 0.456 |
| | avg | **0.397** | 0.414 | 0.429 | 0.436 | 0.475 | 0.462 | **0.417** | **0.424** | 0.392 | 0.411 | 0.397 | 0.414 |
| ECL | 96 | 0.196 | **0.280** | 0.203 | 0.286 | 0.205 | 0.286 | 0.202 | 0.284 | 0.203 | 0.282 | 0.201 | 0.284 |
| | 192 | 0.199 | **0.282** | 0.203 | 0.288 | 0.200 | 0.285 | 0.203 | 0.285 | 0.205 | 0.285 | 0.200 | 0.285 |
| | 336 | 0.215 | **0.296** | 0.222 | 0.307 | 0.216 | 0.299 | 0.218 | 0.299 | 0.221 | 0.300 | 0.216 | 0.301 |
| | 720 | 0.258 | 0.331 | 0.263 | 0.337 | 0.265 | 0.333 | 0.262 | 0.332 | 0.265 | 0.333 | 0.260 | 0.333 |
| | avg | **0.217** | 0.297 | 0.223 | 0.304 | 0.222 | 0.301 | **0.221** | **0.300** | 0.224 | 0.300 | 0.220 | 0.301 |
| Weather | 96 | **0.175** | **0.226** | 0.190 | 0.238 | 0.202 | 0.253 | 0.180 | 0.228 | 0.182 | 0.232 | 0.178 | 0.230 |
| | 192 | **0.225** | 0.270 | 0.239 | 0.279 | 0.248 | 0.289 | 0.227 | **0.268** | 0.235 | 0.274 | 0.229 | 0.271 |
| | 336 | **0.282** | **0.305** | 0.287 | 0.310 | 0.314 | 0.335 | 0.284 | 0.308 | 0.291 | 0.312 | 0.288 | 0.310 |
| | 720 | 0.366 | 0.356 | 0.374 | 0.366 | 0.385 | 0.375 | 0.368 | 0.360 | 0.370 | 0.360 | 0.368 | 0.359 |
| | avg | **0.262** | **0.289** | 0.273 | 0.298 | 0.287 | 0.313 | 0.264 | 0.291 | 0.269 | 0.294 | 0.266 | 0.292 |

