# OpenReview forum: "Kolmogorov-Arnold Networks with Variable Function Basis"
_ICLR.cc/2025/Conference — ICLR 2025 Conference Withdrawn Submission_

### Official Review · Reviewer_wE8n · 2024-10-15

**Soundness:** 3
**Presentation:** 3
**Contribution:** 2
**Rating:** 3
**Confidence:** 5

**Summary:**

The authors proposed KAN augmented with Variable Function Basis, precisely with Bernstein polynomials with degree n as the variable to be inferred like a hyperparameter jointly optimized. Theories concerning Weierstrass approximation theorem justified the choice of polynomials, and Prop 3.2 specifically justified the choice of Bernstein polynomials. Numerical tests were conducted, comparing it to various versions of KANs and MLPs in different tasks.

While the paper is clearly written, the reviewer saw the following challenges.

1. Some comparisons are poor. For example figure 1, you should do a fair comparison using Bsplines with grid points; right now you are comparing approximation between a lower degree polynomial (not piecewise but just one piece) and a higher order one. This is not fair.

2. In the numerics, although a broad number of examples are provided, the referee finds it difficult to be convinced that your setting is better, even judging by the numbers. Please provide more intuitions or more complex examples motivated in detail.

3. Even in the original KANs, you can optimize the spline polynomials k as well. Granted that degree n in B-polynomials might be more robust in terms runge phenomena, that point was missing in the article. I would strongly suggest the authors to motivate more on this since normally one would expect numerical instability as the polynomial degree increases.

4. Can you adaptively choose different degrees n for different layers at least? This will be interesting.

**Strengths:**

See previous section

**Weaknesses:**

See previous section

**Questions:**

Overall, please motivate more why Bernstein polynomials are better and elaborate more on experiments.

---

### Official Review · Reviewer_K5sd · 2024-10-30

**Soundness:** 2
**Presentation:** 2
**Contribution:** 1
**Rating:** 3
**Confidence:** 3

**Summary:**

The paper augments the recently introduced Kolmogorov-Arnold-Network (KAN) architecture by replacing the spline activation function with variety bernstein polynomials (VBn-KAN). The authors argue that this modification increases the expressivity and generalization capabilities compared to vanilla KANs.

**Strengths:**

- The authors extensively motivate the architectural modification of vanilla KANs
- The authors compare their method to various other forms of KANs and other ML models across a variety of tasks, including time series forecasting, image classification as well as general function approximation
- The manuscript is overall well-written

**Weaknesses:**

- Table 1: VBn-KAN improvement over other KAN variants is very marginal
- Table 2 & 3: VBn-KAN is partly outperformed by vanilla KANs
- l. 520 “[...] 0.095 less than the average error [...]” - the authors should try to stay consistent with the use of relative performance improvements/degradations to a baseline. This makes it much easier for the reader to identify the significance of the result (as the authors correctly did when discussing results of Table 1 & 2, for example).
- ll. 532 - 534: “Furthermore, it reduces the influence of data on the model and provides a balanced approach in feature learning by minimizing error while managing computing costs.” - where do the authors show this explicitly in their experiments? To me there is no clear evidence of this in the manuscript.
- ll. 230-231: That the enhanced uniformity contributes to improved model generalization is in this form a claim that is not backed up by evidence (citation or experiments missing).
- Entire subsection 3.2 is filled with claims and theoretical properties of employing VBn-KANs which are not really investigated by experiments.
- While the authors provide a lot of comparison methods/baselines, they do not include SOTA methods for the specific datasets (e.g. for CIFAR-10/100). I understand that the authors just do a comparison of KAN based methods, but it would be useful to state the current SOTA in these benchmarks (based on non-KAN alternatives).

Overall **I vote for reject**, as the paper lacks novelty and impact. For a ICLR conference paper the work lacks substance: The main feat of the paper consists of replacing a activation function in KANs by a, in theory, more expressive alternative. However, the actual gains by this modification are overall only marginal. Indeed, the VBn-KAN does not even consistently outperform vanilla KANs. I also think that there are many claims in the paper that are not backed up by (dedicated) experiments.

Minor details/typos:
- l. 141 typo: “exits” -> “exists”
- ll. 230-231 This sentence needs revision: “This enhances uniformity significantly contributes to improved model generalization.”
- ll 39-40: typo "attempte"
- ll 113-114: typo "interpretabilit"

**Questions:**

- Could the authors include the settings for n for the different curves in the legend of Fig. 1b?
- Could the authors demonstrate how the lower bound for $n$ stated in Theorem 3.3 could be used in practice? Having theoretical guarantees is valuable, but how they play out in practice is also very important. To me the practicality of Theorem 3.3 seems limited or not really investigated, as the authors fall back to (sophisticated) hyperparameter tuning to find the best $n$.
- I do not quite understand Table 4 & 5. What is the n used for VBn-KAN column? Is $n$ optimized by BO and different for each weight in the KAN? If so, could the authors provide some kind of summary statistic to compare VBn-KAN to the fixed $n$ columns?
Also in Table 4 & 5, it seems that for fixed n, the setting $n=1$ generally performs better than $n=10$. Do we observe overfitting for higher values of $n$ here?
- what exactly is $\xi$ in theorem 3.3?
- I assume the results of Table 1-6 are all based on test sets of the individual benchmarks? Please make this clear in e.g. the Table captions.

---

### Official Review · Reviewer_kadZ · 2024-11-03

**Soundness:** 1
**Presentation:** 2
**Contribution:** 2
**Rating:** 3
**Confidence:** 4

**Summary:**

The authors address uniform convergence and variable basis functions in KAN models, and explain in detail how these concepts arise from classical mathematical theory (e.g. Weierstrass theorem). They compare their variable basis model to several existing KAN approaches on three types of machine learning problems (function approximation, time series forecasting, image classification), with favorable results for the new method.

**Strengths:**

The introduction of different types of convergence and their relation to Bernstein polynomials and Weierstrass' theorem demonstrate a good understanding of the fundamental theory for function approximation. The use of Bernstein polynomials in KAN approximations is a novel idea (as far as I know the KAN literature), and the computational results compared to existing KAN approaches seems to favor the new approach.

**Weaknesses:**

* The model should not be compared on image data. The results are worse than the simplest k-NN models (99.04% on MNIST), while the inductive bias from convolutional models outperforms MLPs. The datasets chosen are mere toy models in the computer vision field at this point, and no significant improvements can be reported anymore. If the authors would show how KANs can be used for convolutional filters, this would be an entirely different story.
 * Generally, the authors should incldue results from the state of the art on the given datasets, not just compare to other KAN models. It is ok to be worse in performance than the state of the art, but the tables as they are presented in the paper suggest that the new method is "the best" on the datasets (including the text: l444 "surpassing competing algorithms by substantial margins"), while it really is just "the best" among the KAN methods that are being compared. It is still impressive that the variable basis can outperform the other KAN approaches, but it is misleading to not show the state of the art results. l.448: "Overall, the VBn-KAN model consistently delivers superior or competitive results, underlining its efficacy in various image classification tasks" is a wrong conclusion, because it is not compared to the state of the art. This fact is also not discussed at all.

 * The use of Bayesian optimization is not warranted in the present setting, if in equation 3.5 the only unknown parameter is "n" (a one-dimensional parameter?). In one dimension, MUCH faster algorithms with much better guarantees are available - e.g. a simple line search with bisection has exponentially fast convergence if the function L (Eq. 3.5) is smooth enough.

 * Beyond section 3.3 on "Realization", the authors do not explain at all how they train their KAN model. Which algorithms are used to optimize, how many data points are used, what learning rates, etc?

 * One of the main selling points for KAN models is "interpretability" (however it is defined) of the computed 1D functions. The authors do not discuss this for the computational experiments, but only focus on the uniform convergence property in general.

 * The use of "variety" is not appropriate, I believe (see my question on the topic).
 * small language errors (e.g. l.114 "interpretabilit")

**Questions:**

* L116: what is a "Variety Basis" in the title "Realizing for Variety Basis" (also l121: "variety basis functions", l263: "the need for variety in choosing the appropriate Bn function")? A mathematical variety (e.g. algebraic geometry: set of zeros of polynomial equations) is something very different to what is discussed in this section, no? Even the title of "VBn-KAN" includes "Variety Bernstein Polynomial Function Basis", where I do not understand what "variety" refers to. Maybe I am missing something, or the authors mean "variability" in a more colloquial sense? In l.287 they use "Variable function basis", but then switch back to l304 "variety function basis".

 * Regarding KANs in general: the authors state that (l171) "the network contains no linear weights at all", but then introduce weights w_b and w_s in eq. 3.2 that linearly combine spline and silu functions. Why are these not "linear weights"?

 * Figure 1 is misleading. Why are there not multiple spline functions on the left, with varying degree? This is certainly possible for splines as well - and Bernstein polynomials can also be used as splines... I do not quite understand what the authors want to illustrate here.

---

### Official Review · Reviewer_uFnF · 2024-11-03

**Soundness:** 2
**Presentation:** 2
**Contribution:** 1
**Rating:** 1
**Confidence:** 5

**Summary:**

The paper introduces a novel extension to Kolmogorov-Arnold Networks (KANs) by integrating a variable Bernstein polynomial basis (VBn-KAN). The authors claim this extension enhances the flexibility and robustness of KANs in handling diverse datasets and tasks such as time series forecasting, image classification, and function approximation. While this innovation aligns with the ongoing quest for more interpretable and adaptable neural networks, its practical impact needs clearer justification.

The concept of incorporating a variable function basis, specifically using Bernstein polynomials, is intriguing and builds on established theoretical foundations like the Weierstrass approximation theorem. However, the novelty is somewhat diminished by the lack of a clear demonstration of substantial improvements over existing methods. The results show some benefits, but the advancements seem incremental rather than groundbreaking.

**Strengths:**

The paper introduces the Variety Bernstein Polynomial Function Basis for Kolmogorov-Arnold Networks (VBn-KAN), which is a novel approach in the context of KANs. The authors provide a solid theoretical foundation based on the Weierstrass approximation theorem and propose that Bernstein polynomials can serve as a robust function basis.

The paper demonstrates the model’s flexibility in handling different types of data, including time-series, function approximation, and image classification.

**Weaknesses:**

Unclear Motivation and Practical Importance: The paper does not sufficiently establish why the proposed method is crucial or how it addresses significant gaps in the current state-of-the-art.

Incremental Improvement: The enhancements over existing KANs and other baselines appear marginal. The paper does not convincingly argue that the improvements justify a new architecture.

Comparative Baseline Issues: The comparisons, especially in function approximation, should be more directly against B-splines rather than generic splines, as B-splines are a more appropriate benchmark.

Lack of Robust Evaluation: The experimental setup lacks robustness. More rigorous statistical analysis and ablation studies would strengthen the findings.

Complexity Without Clear Gain: The introduction of a variable basis increases model complexity, but the paper does not adequately demonstrate how this complexity translates into meaningful practical gains.

**Questions:**

1. Comparison Between Spline and Bn Polynomial in Figure 1
The comparison between spline fitting and Bernstein polynomial fitting in Figure 1 is somewhat misleading. The paper mentions KANs using B-splines, not generic splines. B-splines are more flexible because their polynomial degree can be adjusted, which means a simple comparison between spline and Bn polynomial might not be fair. By default, splines are cubic, but B-splines allow for adjustable degrees, which could make the comparison with Bernstein polynomials less biased. The authors should present a more explicit comparison with B-splines of varying degrees, not just generic splines. Additionally, there is no mention of the grid resolution used for the fitting. A grid size could significantly affect the results, and this needs to be clarified for a fair comparison.
Suggestions: 1) Explicitly compare Bernstein polynomials with B-splines of varying degrees
2) Specify the grid resolution used for fitting
3) Discuss how the flexibility of B-splines compares to Bernstein polynomials

2. Uniform Convergence of Bn Polynomials vs. B-Splines
Proposition 3.1 claims that Bernstein polynomials achieve uniform convergence when approximating continuous functions. However, this is not a unique property of Bernstein polynomials. B-splines also achieve uniform convergence when approximating continuous functions. The paper does not sufficiently highlight why Bernstein polynomials are a superior choice over B-splines, especially since both can achieve uniform convergence. The advantages of using Bernstein polynomials should be more clearly articulated and supported by empirical evidence.
Suggestions: 1)Provide a comparative analysis of the uniform convergence properties of Bernstein polynomials versus B-splines
2)Present empirical evidence demonstrating any advantages of Bernstein polynomials
3)Explain their rationale for selecting Bernstein polynomials given the similarities with B-splines


3. Evaluation Metrics in Table 3
The paper uses ADE (Average Displacement Error) as the primary metric for function approximation in Table 3, but it is unclear why this specific metric was chosen over more standard metrics like MSE (Mean Squared Error) or RMSE (Root Mean Squared Error). Were the results using MSE or RMSE unsatisfactory? The paper does not address this. Standard metrics are more commonly used and would make it easier to compare the results with other methods. Without this comparison, the reported performance improvements seem marginal at best, and it is difficult for the reader to grasp the significance of the results.
Suggestions: 1)Explain their rationale for using ADE instead of more standard metrics like MSE or RMSE
2)Provide results using MSE and RMSE in addition to ADE for easier comparison with other methods
3)Discuss how the choice of metric impacts the interpretation of the results and the claimed performance improvements

4. Interpretability and Flexibility
A key contribution of the paper is the introduction of a variable function basis to improve the flexibility of KANs. However, the paper does not convincingly explain how this contributes to improved interpretability in deep learning models. While the authors mention that the variable function basis allows the model to adapt to different data types, there is little discussion on how this enhances the interpretability of the model's decisions. More detailed explanations or examples illustrating the role of function basis variety in improving model interpretability would strengthen this claim.
Suggestion: 1)Provide specific examples demonstrating how the variable function basis enhances model interpretability
2)Discuss the relationship between model flexibility and interpretability in this context
3)Compare the interpretability of their approach to standard KANs and other relevant models

**Details Of Ethics Concerns:**

null

---

### Note · Authors · 2024-11-26

I have read and agree with the venue's withdrawal policy on behalf of myself and my co-authors.